# A Multistage Antigen Complex Epera013 Promotes Efficient and Comprehensive Immune Responses in BALB/c Mice

**DOI:** 10.3390/vaccines11030609

**Published:** 2023-03-07

**Authors:** Chengyu Qian, Xueting Fan, Ruihuan Wang, Bin Cao, Jinjie Yu, Xiuli Luan, Guilian Li, Yi Jiang, Machao Li, Xiuqin Zhao, Danang Fang, Kanglin Wan, Haican Liu, Yongliang Lou

**Affiliations:** 1School of Laboratory Medicine and Life Sciences, Wenzhou Medical University, Wenzhou 325035, China; 2State Key Laboratory for Infectious Disease Prevention and Control, National Institute for Communicable Disease Control and Prevention, Chinese Center for Disease Control and Prevention, Beijing 102206, China; 3School of Public Health, University of South China, Hengyang 421001, China

**Keywords:** *Mycobacterium tuberculosis*, antigen complex, Epera013, immunological evaluation, subunit vaccine

## Abstract

Tuberculosis (TB) remains a serious global health problem. Despite the widespread use of the *Mycobacterium bovis bacillus Calmette-Guerin* (BCG) vaccine, the primary factor for the TB pandemic and deaths is adult TB, which mainly result from endogenous reactivation of latent *Mycobacterium tuberculosis* (MTB) infection. Improved new TB vaccines with eligible safety and long-lasting protective efficacy remains a crucial step toward the prevention and control of TB. In this study, five immunodominant antigens, including three early secreted antigens and two latency associated antigens, were used to construct a single recombinant fusion protein (Epera013f) and a protein mixture (Epera013m). When formulated with aluminum adjuvant, the two subunit vaccines Epera013m and Epera013f were administered to BALB/c mice. The humoral immune responses, cellular responses and MTB growth inhibiting capacity elicited after Epera013m and Epera013f immunization were analyzed. In the present study, we demonstrated that both the Epera013f and Epera013m were capable of inducing a considerable immune response and protective efficacy against H37Rv infection compared with BCG groups. In addition, Epera013f generated a more comprehensive and balanced immune status, including Th1, Th2 and innate immune response, over Epera013f and BCG. The multistage antigen complex Epera013f possesses considerable immunogenicity and protective efficacy against MTB infection ex vivo indicating its potential and promising applications in further TB vaccine development.

## 1. Introduction

As one of the oldest recorded human afflictions, TB is still responsible for considerable human morbidity and mortality worldwide hitherto; the causative agent of TB is MTB. The BCG vaccine is currently the only TB vaccine licensed for human use and is recommended by World Health Organization (WHO) because of its sufficient efficacy in protecting children from tuberculous meningitis and miliary tuberculosis, but the efficacy of BCG in adult pulmonary TB prevention varies and, actually, the BCG vaccination has not reduced the incidence of TB in recent years [1]. A compelling investigation from WHO revealed that a quarter of the world’s population harbors asymptomatically infected MTB, and 10% of these latent tuberculosis infection (LTBI) individuals will develop into active TB as a consequence of endogenous latent MTB reactivation [2]. Taken together with the increasing incidence of MTB/HIV co-infection and multi-drug resistant MTB strains, TB poses one of the greatest challenges for global public health problems [3,4].

Presently, several strategies were used in the development of TB vaccines, which can mainly divide into four types (protein-adjuvant vaccines, live attenuated vaccines, mycobacteria derived/recombinant BCG vaccines and viral vectored vaccines) among which adjuvanted recombinant protein vaccines have drawn increasing attention due to the characteristics of stability in protective efficacy and triggering specific immunological memory without safety concerns of live vaccines [5,6]. According to the Tuberculosis Report of World Health Organization, there were 14 TB vaccine candidates under clinical trials by August 2021, of which five were adjuvanted protein vaccines developed with diversified vaccination strategies, including BCG substitutes, BCG priming-heterologous boosters or therapeutic vaccines aimed at LTBI and TB reactivation [7,8]; the most rapidly advanced subunit vaccines M72/AS01 and H56/IC31 had entered phase 2B clinical trials [9,10]. However, no TB vaccine candidate was proved to result in MTB-specific sterile immunity to prevent LTBI reactivation; accordingly, improved TB vaccines with eligible safety and long-lasting efficacy accompanied with appropriate vaccination strategies remain to be crucial steps toward TB prevention and control.

Well-defined protective antigens were proven to be an essential prerequisite in TB vaccine development [11,12]. Immunodominant antigen candidates for TB vaccine development are mainly extracellular proteins, bacterial membrane and cell wall protein components. Previously, we have evaluated the immunogenicity of several antigens from MTB, both in full-length and truncated modalities (T cell epitope focused regions). The results suggested that the antigens possessed high immunogenicity and may have promising effects on TB vaccine development [13,14,15]. On the basis of these efforts, we designed two multistage antigen complexes Epera013f and Epera013m, which comprised three full-length antigens EsxH, EsxB and Ag85B and two truncated antigens new RipD (nRipD) and new PPE18 (nPPE18), either as a protein mixture (Epera013m) or a fusion protein (Epera013f). EsxH, EsxB and Ag85B are early secreted antigens of MTB and have shown encouraging performance, both as individual antigens and as fusion protein antigens [16,17,18]. Two latency associated antigens—PPE18 and RipD were also selected in the present study. PPE18, a member of the *Mycobacterial* PE/PPE family, was proven to be a crucial virulence factor and associated with intracellular persistence of MTB; the antigen specifically targeted Toll-like receptor 2 (TLR2) and may trigger an anti-inflammatory response by inducing IL-10 production [19]. RipD contains a *NlpC/p60* domain, which exists extensively in *Mycobacterium* species and serves to facilitate the intracellular survival and proliferation of MTB. In-silico epitope prediction technology has been successfully applied to evaluate possible antigenic properties and predict epitopes of proteins in various pathogens [20,21,22]. The frequently used T-cell epitope prediction tools include TEpredict, which is based on the data of HLA allele genotypic frequencies and the online tools of IEDB-AR database (Immune Epitope Database and Analysis Resource). Work by Wang X et al. [14] suggested that a combined utilization of two T-cell epitope prediction tools might improve the accuracy and efficiency of the predicted epitopes and facilitate the advance of vaccine development.

The purpose of the present study was to design and construct two multistage antigen complexes Epera013f and Epera013m containing five antigens EsxH, EsxB, Ag85B, nRipD and nPPE18 and after that to evaluate the immunogenicity and protective efficacy of the two antigen complexes in a mouse model. We report on the prophylactic efficacy of the fusion protein Epera013f, which is able to induce an exquisite and equilibrated Th1, Th2 and trained innate immune response.

## 2. Materials and Methods

### 2.1. T-cell Epitope Prediction

The procedure of T-cell epitope prediction has been described previously [13,14]. The TEpredict software and online tools of IEDB-AR database were utilized for T-cell epitope prediction. Firstly, the amino acid sequences of PPE18 and RipD were obtained from National Centre for Biotechnology Information (NCBI) and submitted to the TEpredict and IEDB-AR, then HLA-A and HLA-B alleles of human major histocompatibility complex (MHC) class I antigen peptide binding were chosen for T-cell epitope prediction (http://tools.iedb.org/mhci/, accessed on 20 September 2019). Common predicted T-cell epitopes were obtained from TEpredict and IEDB-AR; finally the immunogenicity score of every common epitope was predicted using the T-cell pMHC class I immunogenicity predictor from IEDB database (http://tools.iedb.org/immunogenicity/, accessed on 22 September 2019). Sequences enriched with T-Cell epitopes of PPE18 and RipD were chosen and designated as nPPE18 and nRipD, respectively. 

### 2.2. Cloning and Purification of Epera013f and Five Individual Antigen Components of Epera013m

The DNA sequences of *EsxH*, *nPPE18*, *EsxB*, *nRipD* and *Ag85B* were first linked in tandem with peptide linker Gly-Gly-Ser-Gly-Gly and designated as Epera013f, then the DNA sequence of Epera013f was codon-optimized for expression in Escherichia coli (*E. coli*). The optimized sequence was then synthesized with restriction *NdeI* and *XhoI* digest site at 5’ and 3’ end, respectively. After digesting with *NdeI* and *XhoI*, the sequence was ligated into the plasmid pET43.1a. The construction of the recombinant plasmids containing DNA sequences of five individual antigen EsxH, nPPE18, EsxB, nRipD and Ag85B were described as follows: the DNA fragments of the five antigens were firstly amplified from genomic DNA of H37Rv using polymerase chain reaction with *EcoRI* and *HindIII* restriction digest site at 5′ and 3′ end, respectively, followed by *EcoRI* and *HindIII* digestion. The DNA fragments were ligated into the plasmid pET32a. The resulting six recombinant plasmids were confirmed by DNA sequencing. Epera013f and five individual antigens were expressed in *E. coli* host BL21 (DE3), purified using Ni-IDA affinity chromatography (GE Healthcare, Salem, CT, USA) and Q ion-exchange chromatography (GE Healthcare, Salem, CT, USA) on ÄKTA avant25 (GE Healthcare, Salem, CT, USA) and then dialyzed to storage buffer (PBS containing 10% glycerin, pH7.4). The purity of recombinant proteins was determined by sodium dodecyl sulfate polyacrylamide gel electrophoresis (SDS-PAGE). A bicinchoninic acid test (TransGen Biotech, Beijing, China) was used to determine the concentration of the purified protein.

### 2.3. Western Blotting Assay

The presence of five components within Epera013m was confirmed by immunoblotting with mouse polyclonal serum raised against EsxH, nPPE18, EsxB, nRipD and Ag85B, respectively (1:1000), then followed by horseradish peroxidase (HRP)-conjugated anti-mouse IgG (1:5000; Biodragon, Beijing, China) incubation. The absence of *E. coli* intrinsic constituent was confirmed by immunoblotting with mouse polyclonal serum raised against Epera013f (1:1000) and followed by HRP-conjugated anti-mouse IgG (1:5000; Biodragon, Beijing, China) incubation. Electro-chemiluminescence method was used to visualize the blots (Thermo Fisher, Waltham, MA, USA)

### 2.4. Culture Condition and Preparation of Mycobacterium Strains

For vaccination and ex vivo MTB infection, BCG-Pasteur and H37Rv (ATCC25618) was grown on Lowenstein–Jensen medium at 37 °C for approximately 4 weeks before washing with aseptic PBS and ultrasonic dispersion. Bacterial suspensions were measured with McNamara turbidimetric tube assay, then diluted to 5 × 10^6^ colony-forming unit (CFU) per milliliter (BCG-Pasteur, for vaccination) and 500 CFU per milliliter (H37Rv, for ex vivo infection). Procedures involving H37Rv were done at biological safety third-level laboratory (BSL-3).

### 2.5. Animals and Immunization Strategy

The detailed vaccination schedules were exhibited in Figure 1. Six-week-old female BALB/c mice were obtained from Beijing HFK Bioscience Co.127 Ltd. (Beijing, China) and randomly classified into five groups (*n* = 6); animals were maintained under specific pathogen-free conditions and were treated in accordance with the regulations and guidelines of Animal Care and Welfare Committee of NIFDC. The vaccines were prepared with 50 μg Epera013f or 50 μg Epera013m incorporating five antigens in the same mass (minus the molecular weight of the tags within pET32a plasmid while calculating the dose of the five proteins) mixed with 50 μL aluminum adjuvant (Thermo Fisher, Waltham, MA, USA) at the volume ratio 3:1, respectively. All mice were vaccinated subcutaneously three times with 10 days apart. The positive control mice were immunized with 1 × 10^6^ CFU of BCG; the blank and negative control groups were given 200 μL PBS and 200 μL adjuvant (50 μL aluminum adjuvant mixed with 150 μL PBS), respectively. The mice of all groups were sacrificed for downstream assessments 10 days after the last vaccination.

### 2.6. Humoral Immunogenicity Evaluation 

Enzyme-linked immunosorbent assay (ELISA) was applied to determine serum antigen specific IgG and the isotypes IgG1 and IgG2a. Under aseptic conditions, the blood of the mice was collected by retro-orbital puncture 10 days after the last immunization and stranded at room temperature for 2 h, then the serum was separated by centrifuging at 4000 rpm for 10 min. The 96-well ELISA plates were coated overnight with either Epera013f, Epera013m or BCG whole bacteria lysate (BCG antigens) at the quantity of 200 ng/well diluted in coating buffer (50 mM sodium carbonate buffer, pH 9.6). The plates were washed five times with PBST (PBS containing 0.5‰ Tween 20) and blocked with 200 μL blocking buffer (PBS containing 5% skim milk) at 37 °C for 2 h. After five times of plate washing with PBST, 100 μL/well double-diluted serum samples were added to the plates coated with corresponding antigens in triplicate and incubated for 1 h at 37 °C. After that, the plates were washed 5 times with PBST and 100 μL HRP-conjugated anti-mouse IgG, IgG1 or IgG2a polyclonal antibody (1:5000; Biodragon, Beijing, China) was added per well, followed by incubation at 37 °C for 1 h. After 5 times of plate washing, 100 μL tetramethylbenzidine (TMB) substrate solution (TIAGEN Biotech, Beijing, China) per well was added and incubated at 37 °C for 15 min. Finally, 100 μL terminate buffer (2M H_2_SO_4_) was added to each well and measured the absorbance at 450 nm using microplate reader (BIO-RAD Laboratories, Hercules, CA, USA). Serum from the PBS group was served as negative control.

### 2.7. Lymphocyte Preparation

After the mice were sacrificed by cervical dislocation followed by retro-orbital puncture blood collection, spleens were separated under sterile conditions and grinded for splenocytes separation using mouse lymphocyte separation medium (DAKEWE, Shenzhen, China, 7211011) and density gradient centrifugation. After erythrocyte lysis (Solarbio LIFESCIENCES, Beijing, China), single cell suspension of mice splenocytes were adjusted to 2 × 10^6^ cells per milliliter with antibiotic-free complete medium RPMI1640 (Gibco, Waltham, MA, USA) containing 10% heat-inactivated FBS (Gibco, Waltham, MA, USA) for downstream assays. A pre-experiment was performed to acquire the appropriate dose of the antigens used for splenocytes restimulation without cytotoxicity [23]. 

### 2.8. Antigen Specific Cytokines Determination 

First, enzyme-linked immunospot (ELISPOT) assays were applied to examine the interferon-γ (IFN-γ) and interleukin-4 (IL-4) secreting cells restimulated with Epera013f, Epera013m or BCG antigens (BD Biosciences, Franklin Lakes, NJ, USA). Both splenocytes of the PBS and adjuvant groups were stimulated with the above three antigens, respectively. The results from the PBS groups served as blank controls while that from the adjuvant groups served as negative controls. All assays were performed in triplicate. According to the manufacturer’s instructions, in the ELISPOT plates which had been coated with IFN-γ or IL-4 specific capture antibody following blocking the free binding sites, 2 × 10^5^ splenocytes per well were seeded and stimulated with 2 μg corresponding antigens or BCG antigens. After 24 h of incubation in 37 °C, 5% CO_2_ and humidified incubator, the plates were washed and added with biotin-conjugated detection antibody and incubated at room temperature for 2 h. After the plates were washed and added with streptavidin labeled HRP and incubated at room temperature for 1 h and washed again, 100 μL 3-amino-9-ethylcarbazole (AEC) substrate solution (BD Biosciences, Franklin Lakes, NJ, USA) was added and incubated at room temperature for distinct spot formation, finally the reaction was terminated by deionized water rinse. The plates were dried, and spot-forming cells (SFCs) were enumerated using ELISPOT plate reader (AID EliSpot Reader System, Straßberg, Germany). Second, Luminex multiplex cytokine assays were also applied to detect secretory cytokines of IL-2, IL-4, IL-6, IL-10, IL-12p70, IL-17A, IFN-γ, tumor necrosis factor-α(TNF-α) and granulocyte-macrophage colony stimulating factor (GM-CSF) according to the instructions of manufacturer; splenocytes from the PBS group were used as negative control. Briefly, 2 × 10^5^ splenocytes were co-cultured with 10 μg of the corresponding antigens or BCG antigens and incubated at 37 °C, 5% CO_2_ and humidified condition for 24 h. After that the supernatants were collected and secretory profiles for the above cytokines were determined in triplicate using custom Luminex Mouse Magnetic Assay (R&D Systems, Minneapolis, MN, USA). 

### 2.9. Ex Vivo Mycobacterial Growth Inhibition Assays

Assays to determine the inhibition of MTB growth evaluate both T cell and innate immune cell functions [24]. Mycobacterial growth inhibition assays (MGIAs) have been proposed as a simple and unbiased tool to evaluate the protective efficacy of TB vaccines in vitro [24,25]. A total of 2 × 10^6^ splenocytes within 1.5 mL antibiotic-free complete medium supplemented with 10 mM Hepes (Gibco, Waltham, MA, USA) and 2 mM L-Glutamine (Gibco, Waltham, MA, USA) were seeded in triplicate in 24-well tissue culture plates; 50 CFU ultrasonic dispersed H37Rv per well were added and incubated at 37 °C, 5% CO_2_ and humidified condition for 96 h. After incubation, the cells were pelleted through a 12,000 rpm, 10 min centrifugation and lysed by vortex in 500 μL sterile water. Finally, the sediments were resuspended in 500 μL sterile water and 50 μL suspension was inoculated on 7H10 medium supplement with 10% OADC (BD Biosciences, Franklin Lakes, NJ, USA) and incubate at 37 °C for three weeks. Number of bacteria colonies were enumerated and the data were presented as total number of CFU per milliliter. Splenocytes from the PBS and adjuvant groups served as blank and negative control, respectively. 

### 2.10. Statistical Analysis

The statistical analysis was performed using IBM SPSS statistical software package (version 26.0, IBM Corp, New York, NY, USA). One-way analysis of variance (ANOVA) was used to compare the results between groups in ELISA assays, ELISPOT assays and MGIAs, and LSD (Least Significant Difference) multiple comparison tests was used to perform two-by-two comparisons as applicable. Nonparametric Kruskal–Wallis test was performed for the comparison of secretory cytokines from the Luminex multiplex cytokine assays. Experimental data were presented as the mean ± standard error. *p* value < 0.05 was considered statistically significant.

## 3. Results

### 3.1. T-cell Epitope Prediction

The distribution of all predicted epitopes with satisfactory immunogenicity scores in PPE18 and RipD were located in five domains (Figure 2A(D1–D5)) and six domains (Figure 2B(D1–D6)), respectively. In the present study, we chose amino acid sequences of PPE18 and RipD enriched with T-Cell epitopes and designated as nPPE18 (201-300 aa) and nRipD (37-184 aa), respectively. Detailed position information of two truncated antigens were shown in Table 1.

### 3.2. Expression and Purification of the Recombinant Proteins

An optimized 2364-bp DNA fragment (Epera013f) was inserted into the plasmid pET43.1a; the DNA sequences of five individual antigens EsxH (291 bp), nPPE18 (300 bp), EsxB (303 bp), nRipD (444 bp) and Ag85B (978 bp) were inserted into the plasmid pET32a (Figure 3B), then the recombinant plasmids were confirmed by DNA sequencing. The Epera013f recombinant protein composed five MTB antigens EsxH, nPPE18, EsxB, nRipD and Ag85B linked in tandem (Figure 3A). SDS-PAGE analysis of the fusion protein Epera013f and five individual antigen components of Epera013m showed a single major band in line with the expected molecular weights of 82.4 kDa, 28.4 kD, 29.6 kD, 28.8 kD, 34.9 kD and 52.6 kD, respectively (Figure 3C). The identification of the five constituents of Epera013f were also confirmed by Western blotting (Figure 3D). Epera013f was recognized by polyclonal serum raised against each of the five antigens confirming the presence of five antigens within the fusion protein. None of *E. coli* component was found in purified Epera013f antigen by Western blotting (Figure 3E).

### 3.3. Robust Humoral Immune Responses Were Induced by Epera013f and Epera013m 

To determine the humoral immune response irritation capacity of the Epera013f and Epera013m, ELISA assays were performed to measure the magnitude of serum antigen specific IgG and the isotypes IgG1 and IgG2a in the immunized BALB/c mice 10 days after the last vaccination. The results showed that high levels of serum IgG, IgG1 and IgG2a in the Epera013f, Epera013m and BCG groups (Figure 4A), whilst low absorbance of the PBS and adjuvant groups were measured. Epera013m immunized mice showed the highest serum IgG level as compared with Epera013f and BCG (both *p* values were <0.01); the titers of IgG1 and IgG2a in the two protein-adjuvanted groups were both higher than that of the BCG group (*p* values were <0.01 and <0.05, respectively). As shown in Figure 4B, the IgG1/IgG2a ratios of the Epera013f, Epera013m and BCG groups were all floating around one, and the ratio of the BCG group was lower than that of the Epera013f and Epera013m groups (*p* values were <0.05 and <0.01, respectively), suggesting a moderate extent of Th1 immune response in BCG immunized mice and Th1–Th2 balanced immune response in Epera013f and Epera013m immunized mice. 

### 3.4. Th1, Th2 and Innate Immune Response Equilibrated Immune Status Was Generated by Epera013f

We conducted experiments to characterize the antigen-specific cytokine responses induced by Epera013f, Epera013m and BCG in the immunized BALB/c mice by means of ELISPOT assays and Luminex multiplex cytokine assays. 

First, antigen-specific IFN-γ and IL-4 were measured 10 days after the last immunization. The results displayed that all five individual antigens were able to induce an increased number of IFN-γ and IL-4 secreting cells in the Epera013f and Epera013m groups (Figure 5A,B). Meanwhile, the IFN-γ and IL-4 secreting cells in the protein subunit and BCG groups were significantly higher than that of the PBS and adjuvant groups (Figure 5C,D, both *p* values were <0.01). IFN-γ secreting cells in the Epera013f group showed no significant difference in comparison with the BCG group (*p* > 0.05) but were higher than that of the Epera013m group (*p* < 0.05), whereas cells secreting IL-4 in the Epera013f group were higher than that of the Epera013m and BCG groups (both *p* values were <0.05). The IFN-γ: IL-4 ratios of the Epera013f and Epera013m groups were both approximately one, indicating a Th1–Th2 balanced immune response induced by Epera013f and Epera013m.

Second, secretory profiles for nine cytokines in the culture supernatant of mice splenocytes restimulated with Epera013f, Epera013m or BCG antigens were measured using Luminex assays. These cytokines included Th1 cytokines IFN-γ, IL-2, IL-12p70 and TNF-α and Th2 cytokines IL-4, IL-6 and IL-10, and innate immunity associated cytokines GM-CSF and IL-17A. Splenocytes of the PBS group were used as negative control (data not shown). As shown in Figure 6, a significant increase in the productions of IFN-γ and TNF-α was observed in Epera013f immunized mice compared with the Epera013m and BCG groups (both *p* values were < 0.05); the levels of IL-12p70 in mice immunized with the two protein subunit vaccines were both higher than that of the BCG group (both *p* values were <0.05). In terms of the Th2 cytokines, the secretion levels of IL-4, IL-6 and IL-10 in the Epera013f group revealed a significant increase compared to that of the Epera013m and BCG groups (*p* values were all <0.05). With regard to the innate immune response, Epera013f vaccinated mice exhibited potentiated secretion capacity of GM-CSF compared with the Epera013m and BCG groups (both *p* values were <0.05). In addition, although the difference fell short of statistical significance, Epera013f showed the trend to induce higher IL-2 and IL-17A secretion levels in comparison with Epera013m and BCG. These results suggested Epera013f could induce a more comprehensive and equilibrated immune response comprising of cellular, humoral and trained innate immune response among the three studied groups.

### 3.5. Mycobacterial Growth Inhibiting Capacity of Murine Splenocytes

MGIAs were applied to evaluate the inhibitory efficacy of the splenocytes from two subunit vaccines and BCG immunized groups against H37Rv growth; splenocytes from the PBS and adjuvant groups were used as blank and negative controls, respectively. Macrophages, neutrophils and T cells play important roles in the phagocytosis and clearance of MTB, and the protective efficacy of the TB vaccine mainly depends on how well these cells are activated, which was reflected as the reduction in the number of colony forming units in MGIA assays. After co-culture of the murine splenocytes and H37Rv for 96 h, a significantly reduced number of bacteria colonies was observed in the Epera013f, Epera013m and BCG groups in comparison with the PBS and adjuvant groups (both *p* values were <0.01, Figure 7), whilst no significance was found among the three studied groups though the BCG group showed a higher trend in inhibiting effect.

## 4. Discussion

TB remains a serious global health problem despite the wide use of the BCG vaccine, which was reported to have unreliable protective efficacy against adult pulmonary TB, and consequently, TB vaccine development remains an ongoing field of research for TB prevention and control. Being different from many other pathogens, infection with MTB does not necessarily lead to active TB disease. MTB is capable of surviving in an intracellular habitat for years and confers LTBI [26]. MTB exhibits dynamic conditions in response to the host environment, while suffering from hypoxia, immune pressure and nutrient starvation in the stationary stage, the transcriptional profile of MTB is distinct from that in the early infection stage [27,28,29], which may cause MTB escaping from the pre-established antigen specific immunological memory and may be one of the reasons why the protective efficacy of BCG declines in adulthood. 

Controlling the spread of TB has become an urgent public health priority, which may be achieved by developing advanced TB vaccines or by improving the long-term protection provided by the BCG vaccine with prime/boost strategies. In the recent pipeline of TB vaccine development, recombinant fusion proteins with multiple antigens were proven to induce an increased protective immune response against MTB [1,6,17], and the antigen selection tactic has transformed from the combination of early secreted antigens into multistage antigen compounds with latency associated antigens comprised [17,30]. The development of H56/IC31 and ID93/GLA-SE were based on the prominent virulence factors of MTB with latency associated antigens Rv2660c and Rv1813 incorporated, respectively. The two vaccine candidates both exhibited inspiring clinical results as therapeutic vaccines. In this study we selected five well-defined protective antigens of MTB, including members of the virulence factor families Esx (EsxH and EsxB), early secreted antigen (Ag85B), intracellular persistence associated antigen (RipD) and latency associated antigen (PPE18), and constructed two multistage antigen complexes either as a protein mixture or a single recombinant fusion protein, termed as Epera013f and Epera013m, respectively. We demonstrated that high levels of humoral immunity response and robust inhibiting capacity for mycobacterial growth were generated by Epera013f and Epera013m, whilst only Epera013f exhibited the capacity for inducing polyfunctional T cell responses characterized by the production of balanced antigen-specific cytokine profiles associated with Th1, Th2 and trained innate immune responses. 

Previous literature showed that protective antimycobacterial immune responses depended critically on the activation of the Th1 axis, including IL-12, IFN-γ, IL-2, TNF-α, IL-23 and other Th1 associated cytokines [31]. Being a pivotal Th1 cytokine associated with MTB infection control, IFN-γ was endorsed by WHO to be an applicable immunological biomarker and plays an important role in the development of MTB diagnostic tools and TB vaccines [32]. Our results showed that Epera013f vaccination could induce a strong Th1 immune response encompassing all five antigen components. The IFN-γ secreting capacity in response to the Epera013f restimulation was the highest among that of the Epera013f and BCG restimulation. The bioactive IL-12 (IL-12p70) is a heterodimeric cytokine composed of an inducible IL-12p40 subunit and a constitutively produced IL-12p35 subunit [33]. A recent study manifested that patients with deficiencies in IL-12Rβ1 or IL-12p40 expression were predisposed to develop progressive TB between the ages of 2.5 and 12 and suffer from both TB and mycobacterial disease caused by MTB or weakly virulent BCG and environmental Mycobacterium [34]. In our study, the secretion levels of IL-12 in Epera013f and Epera013m groups were higher than that in the BCG group, implying the activation of the IL-12/IFN-γ axis and may induce a more comprehensive immune response against MTB. Convincing evidence revealed that polyfunctional T cells with tripartite capacity for producing IFN-γ, TNF-α and IL-2 were implicated in the induction of sustained vaccine-induced immunological memory responses [35]. Results of TNF-α secretion in the Epera013f group were higher than that in Epera013m and BCG groups; nevertheless, our results indicated that IL-2 might not confer the protective immune response because of the limited expression efficiency both in the protein subunit and BCG immunized mice.

The roles of the Th2 immune response and the effects of antibodies in host antimycobacterial are subject to some controversy as MTB is an intracellular pathogen. Studies on B cell–deficient mice infected with MTB have yielded various results, with reports of delayed pathological progression of no apparent effect and of a diminished immune response. The differential effects of humoral immune response on the anti-TB response toward immune protection or pathogenesis may depend on the host immunity, MTB infection stage and host/bacilli genetic background [31]. More recently, investigators have identified positive effects of activated B cells and antibody in granuloma formation of nonhuman primates infected with MTB, indicating that humoral immune response was likely to participate in the orchestration of local immune defense against MTB and the immunomodulation of MTB infected hosts [36,37]. IL-4 and IL-6 are important humoral immunity associated cytokines secreted primarily by Th2 cells and macrophages. Results of ELISPOT assays in our study indicated that both Epera013f and Epera013m vaccination resulted in an adequate Th2 immune response incorporating all five antigen components. The IL-4 secreting capacity of splenocytes in the Epera013f group was higher than the Epera013f and BCG groups. In addition, higher levels of IL-6 expression were induced by Epera013f immunization compared with Epera013m and BCG. IL-10 is an important immunomodulatory factor; recent studies reported a positive effect of IL-10 in host antimycobacterial responses [38,39]. Our results indicated a significantly higher secretory ability of IL-10 in the Epera013f group. IgG, IgG1 and IgG2a were confirmed to participate in the antimycobacterial responses [40], and antigen specific IgG is the dominant effective factor for humoral immunity. The two isotypes IgG1 and IgG2a are Th1-type and Th2-type antibodies, respectively. In the present study, a pronounced higher level of serum IgG, IgG1 and IgG2a were observed in Epera013f, Epera013m and BCG vaccinated mice, amongst which Epera013m induced the highest IgG titers, whilst the titers of IgG1 and IgG2a in the protein subunit immunized groups were both higher than that of the BCG group. The IgG1/IgG2a ratios of the Epera013f, Epera013m and BCG groups were approximately one, and the ratio of the BCG group was lower than that of the Epera013f and Epera013m groups, suggesting a moderate extent of Th1 biased immune response in BCG immunized mice and a Th1–Th2 balanced immune response in Epera013f and Epera013m immunized mice.

Protection against MTB induced by BCG vaccination could be elicited partially through the education of the innate immune system, which is capable of mounting a more effective immune response against MTB or even unrelated pathogens [41,42]. Multiple pathways of recognition and signaling of the innate immune response against MTB infection have been well characterized, and the importance of these responses in the TB control are identified by the early mortality and high bacterial burden observed in related gene-deleted mice [38]. GM-CSF and IL-17A are both important cytokines participating in the host innate immunity against MTB [43,44]. In this study, the secretion level of GM-CSF in the Epera013f group was obviously higher than that of the Epera013m and BCG groups; slightly increased IL-17A production was also found in the Epera013f, Epera013m and BCG groups compared with the PBS or adjuvant group. While there was no significant difference among the three groups, it was generally observed that Epera013f induced a slightly higher level of IL-17A secretion as compared with Epera013m and BCG. 

A tripartite requirement for cell-mediated immunity and humoral immunity, as well as trained innate immunity in the protective immune response against MTB, is increasingly recognized presently despite the demonstration of specific T cells as the pillar of the acquired immune response against MTB infection [41,45]. Results of Rodo MJ et al. [46] revealed that although strong Th1-biased immune responses were generated by most TB vaccines hitherto, the lack of diversity in immunological responses may be the reason why BCG and the six TB vaccine candidates did not demonstrate adequate antimycobacterial capacity. In the present study, two key immune response characteristics—antigen specific cytokine expression profiles of memory T cells and immune response magnitude induced by Epera013f and Epera013m, were evaluated. The results revealed that robust and Th1-Th2 balanced cytokine profiles were induced by both Epera013f and Epera013m, but Epera013f exhibited a repertoire to generate a more comprehensive immune response. Intriguingly, there were differences between MGIAs results and antibody/cytokine characteristics: limited cytokines were induced by BCG vaccination, but results of MGIA assays revealed an equivalent mycobacterial growth inhibition among the three groups, which could attribute to the mechanisms of innate and adaptive immunity. The host antimycobacterial response exhibits the characteristics of complexity and integrality, whereas results of serum antibodies and cytokine profiles remain individual and separated, as a result comprehensive protective ability of vaccines against MTB infection reflects mainly in vivo MTB challenge assays or partially by MGIAs rather than the results of immune biomarkers. In consideration of the intrinsic capacity of BCG in protecting MTB primary infection, results of MGIA assays indicated that Epera013f and Epera013m would provide considerable protective efficacy against H37Rv infection.

In consideration of the dynamic conditions of MTB in different infection stages and differences of experimental results due to several animal models [17,47], the Epera013 antigen complex warrants further preclinical studies to evaluate the immunogenicity in different animal models, including latent infection, reactivation and BCG vaccination boosting in various mouse and guinea pig models.

## 5. Conclusions

Our present study revealed that the multistage antigen complex Epera013f was capable of generating a substantial and extensive antimycobacterial immune response providing support for further development in different animal models with optimized vaccination strategies. Furthermore, Epera013f is a potential and promising antigen compound and could be used as a reference antigen in new TB vaccine development.

## Figures and Tables

**Figure 1 vaccines-11-00609-f001:**
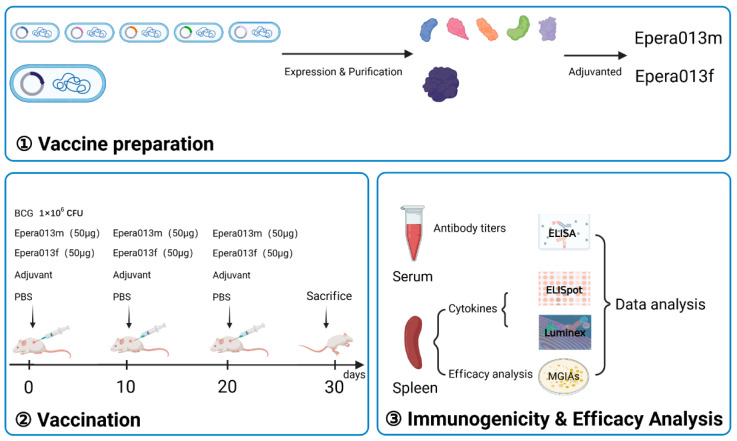
Schematic diagram of experimental design and procedures of the immunogenicity and efficacy evaluation of Epera013, Epera013m and BCG. CFU, colony forming unit; ELISA, enzyme linked immunosorbent assay; Elispot, enzyme linked immunospot assay; MGIAs, mycobacterium growth inhibition assays.

**Figure 2 vaccines-11-00609-f002:**
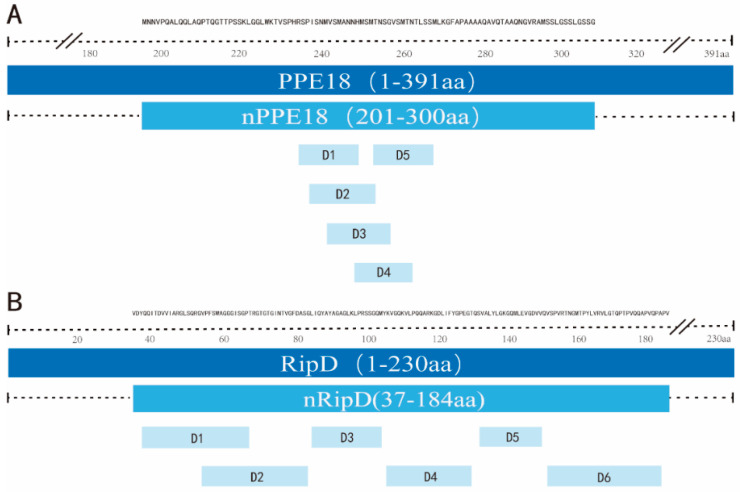
Distribution of T-cell epitope-rich domains in genes of PPE18 (**A**) and RipD (**B**). Abbreviation: aa, amino acids.

**Figure 3 vaccines-11-00609-f003:**
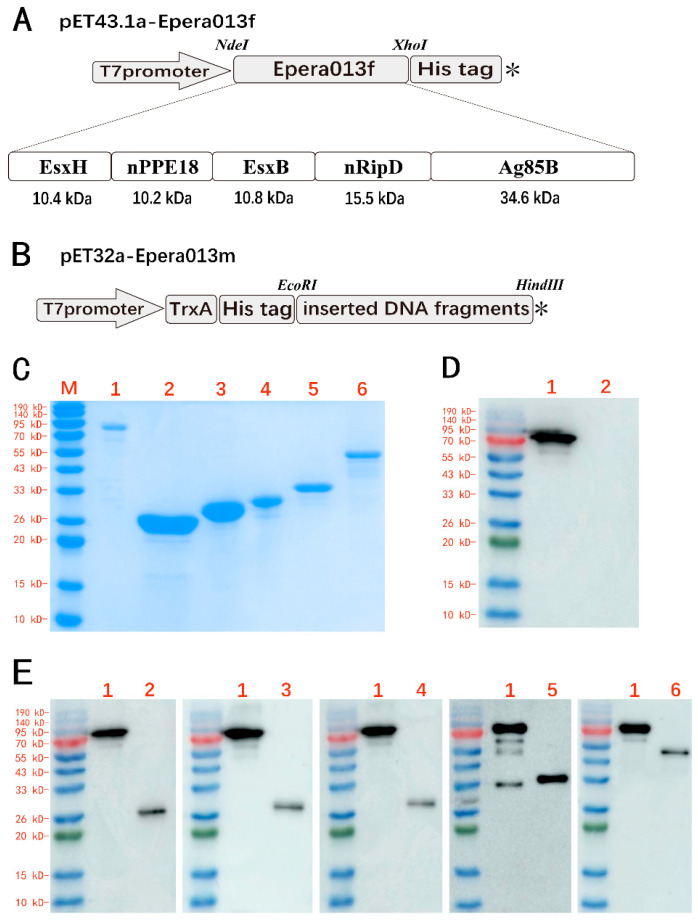
Epera013f recombinant protein construction and characterization. (**A**) The structural diagram of the recombinant prokaryotic expression plasmid pET43.1a-Epera013f. (**B**) The structural diagram of the recombinant plasmid constituted Epera013m. (**C**) SDS-PAGE analysis of purified Epera013f protein and five single individual proteins of Epera013m. Lane M, protein molecular weight marker; lane 1, Epera013f; lane 2, EsxH; lane 3, nPPE18; lane 4, EsxB; lane 5, nRipD; lane 6, Ag85B. (**D**) Immunoblot of Epera013f with mouse polyclonal serum raised against EsxH, nPPE18, EsxB, nRipD and Ag85B, respectively. Lane 1, Epera013f; lane 2, EsxH; lane 3, nPPE18; lane 4, EsxB; lane 5, nRipD; lane 6, Ag85B. (**E**) Immunoblot of *E. coli* intrinsic constituent in purified Epera013f protein. Line 1, Epera013f; line 2, *E. coli* whole cell lysate. M, protein molecular weight marker. (✻, termination codon).

**Figure 4 vaccines-11-00609-f004:**
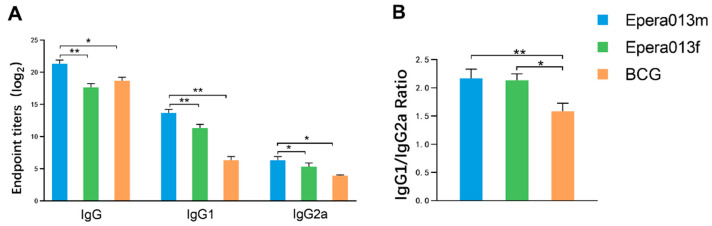
Comparison of serum antibody levels of immunized mice among the Epera013f, Epera013m and BCG groups (*n* = 6 for each group, mean values ± S.E.M.). (**A**) Endpoint titers of serum IgG and two isotypes IgG1 and IgG2a in Epera013f, Epera013m and BCG groups. (**B**) Serum IgG1/IgG2a ratio of immunized mice in Epera013f, Epera013m and BCG groups. Values were from 3 independent experiments. One-way ANOVA followed by LSD (Least Significant Difference) multiple comparison test was performed for the comparison among Epera013f, Epera013m and BCG groups (* *p* < 0.05, ** *p* < 0.01).

**Figure 5 vaccines-11-00609-f005:**
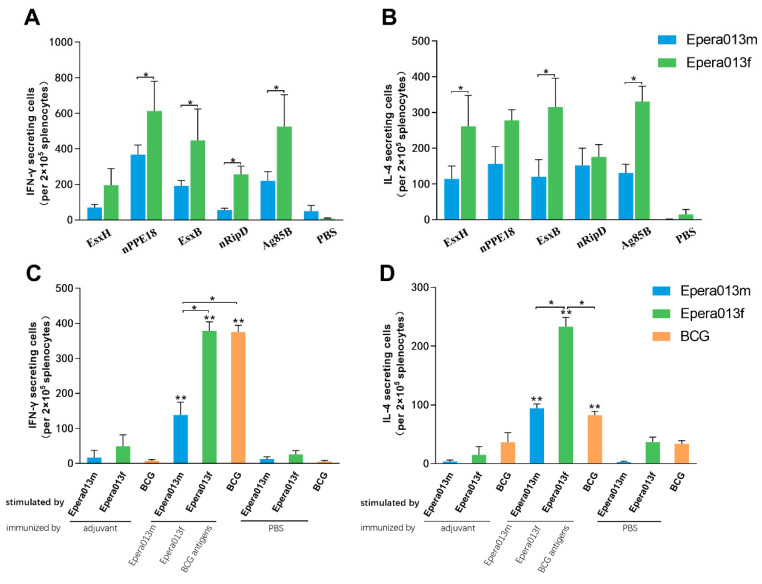
Results analysis of antigen-specific IFN-γ and IL-4 secretion levels of immunized mice among Epera013f, Epera013m and BCG groups (*n* = 6 for each group, mean values ± S.E.M.). (**A**) IFN-γ and (**B**) IL-4 secreting cells in response to the five individual protein restimulation. (**C**) IFN-γ and (**D**) IL-4 secreting cells in response to Epera013f, Epera013m and BCG antigens restimulation. Significant statistical differences between adjuvanted antigen/BCG groups and PBS/adjuvant control groups stimulated with corresponding antigens or BCG antigens were represented by asterisks above the error bars. Significant statistical differences among Epera013f, Epera013m and BCG groups were presented by capped lines with asterisks. Values were from three independent experiments. Statistical analysis of the numbers of IFN-γ and IL-4 secreting cells between groups were performed with one-way ANOVA and LSD (Least Significant Difference) multiple comparison test. (* *p* < 0.05, ** *p* < 0.01). IFN-γ, interferon-γ; IL, interleukin.

**Figure 6 vaccines-11-00609-f006:**
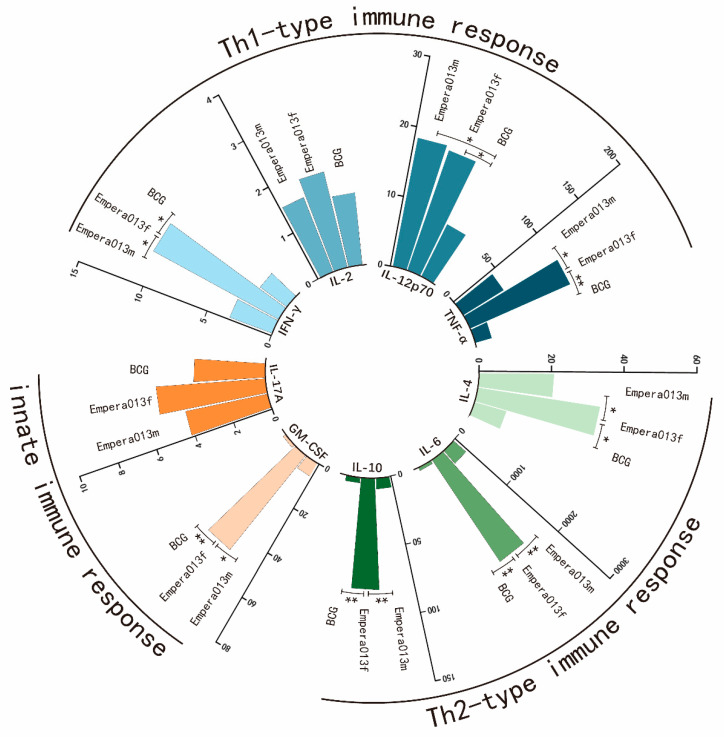
Antigen-specific cytokine secretion levels of murine splenocytes stimulated with corresponding antigens or BCG antigens (*n* = 6 for each group, mean values ± S.E.M.). IL-2, IL-4, IL-6, IL-10, IL-12p70, IL-17A, IFN-γ, TNF-α and GM-CSF were measured using Luminex multiplex cytokine assays. Values were from three independent experiments. Nonparametric Kruskal–Wallis test was performed for the comparison among Epera013f, Epera013m and BCG groups (* *p* < 0.05, ** *p* < 0.01). IFN-γ, interferon-γ; IL, interleukin; TNF, tumor necrosis factor; GM-CSF, granulocyte-macrophage colony stimulating factor.

**Figure 7 vaccines-11-00609-f007:**
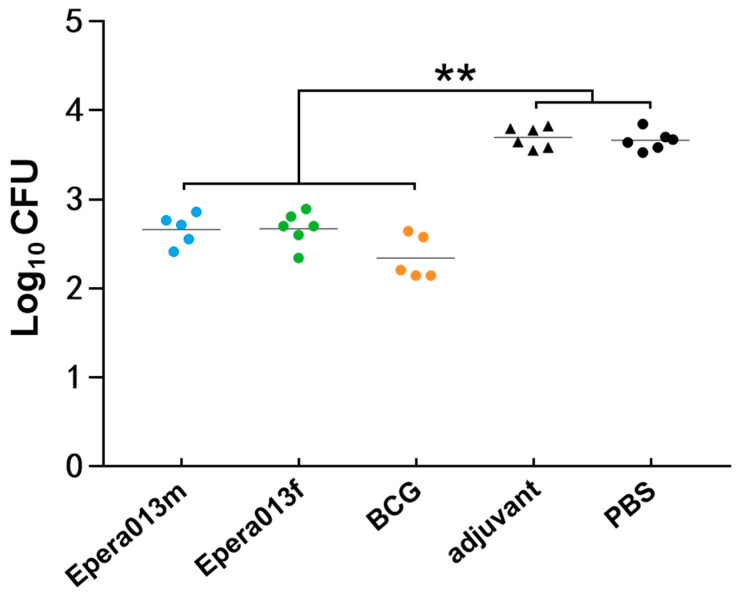
Efficacy analysis of splenocytes from Epera013f, Epera013m and BCG immunized mice in the inhibition of MTB growth ex vivo (*n* = 6 for each group, mean values ± S.E.M.). Values were from three independent experiments. One-way ANOVA followed by LSD (Least Significant Difference) multiple comparison test was performed for the comparison among Epera013f, Epera013m, BCG groups and blank/negative controls (PBS/adjuvant groups) (** *p* < 0.01). CFU, colony-forming unit. Triangles and circles correspond to the horizontal coordinates: blue circles refer to Epera013m group, green circles refer to Epera013f group, yellow circles refer to BCG group, black triangles refer to adjuvant group, black cirles refer to PBS group.

**Table 1 vaccines-11-00609-t001:** Nucleotide and amino acid positions of nPPE18 and nRipD enriched with T-cell epitopes.

		Start at	End at	Length
nPPE18	Nucleotide position	601	900	300 bp
	Amino acid position	201	300	100 aa
nRipD	Nucleotide position	109	552	444 bp
	Amino acid position	37	184	148 aa

Abbreviations: bp, base pairs; aa, amino acids.

## Data Availability

Not applicable.

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
