# Peer review of "A Multistage Antigen Complex Epera013 Promotes Efficient and Comprehensive Immune Responses in BALB/c Mice"

_vaccines, 2023, doi:10.3390/vaccines11030609_

Round 1

Reviewer 1 Report

The manuscript provides useful information on the antimycobacterial immune response. The paper is well introduced.

Title reflects complexity and main objectives of the paper.

Please re-write the last paragraph of the Introduction to highlight the scientific question of thecurrent study.

General comments: Strenghts and limitations: Strenghts: good philosophy and methods, systematic research, innovative and applicable research. Results were properly analyzed and t generally well-written. 

Limitations: lack of hypothesis,  dicussion is not well explorated mainly when authors report  about MTB infection stage and  host/bacilli genetic background. Other point is in line 624 (Rodo MJ et al. ) there is no reference numbering. 

Line 306: Add information on how to interpret the results related to the ex vivo mycobacterial growth inhibiting assays. And also make

these clear in the tables where they are mentioned, and when discussing the results mention any statistically significant data obtained.

Lines 644-649: For your conclusions, I would make it stringer by adding a final sentence about the relevance and impact (e.g. economic and health) of your results.

Author Response

Response to Reviewer 1 Comments

Point 1: Please re-write the last paragraph of the Introduction to highlight the scientific question of the current study.

Response 1: Thank you for your suggestion, the last paragraph of the Introduction has been rewritten (lines 85-90).

Point 2: General comments: Strengths and limitations: Strengths: good philosophy and methods, systematic research, innovative and applicable research. Results were properly analyzed and generally well-written. Limitations: lack of hypothesis, discussion is not well explorated mainly when authors report about MTB infection stage and host/bacilli genetic background. Other point is in line 624 (Rodo MJ et al.) there is no reference numbering.  

Response 2: Thank you for your general comments, aiming at the limitations you mentioned, we have made corresponding improvement (lines 650-654).

The reference (Rodo MJ et al.) has been added (line 630).

Point 3: Line 306: Add information on how to interpret the results related to the ex vivo mycobacterial growth inhibiting assays. And also make these clear in the tables where they are mentioned, and when discussing the results mention any statistically significant data obtained.

Response 3: Thank you for your advice, the information about MGIAs have been added both in methods and results sections, and all the statistically significant results have been mentioned (lines 211-212, 308-317).

Point 4: Lines 644-649: For your conclusions, I would make it stringer by adding a final sentence about the relevance and impact (e.g. economic and health) of your results.

Response 4: Thank you for your suggestion, the conclusion part has been revised in accordance to the suggestion (lines 657-661).

Reviewer 2 Report

The present study may be strengthened by addressing the following queries:

In the introduction section, lines 30, authors should expand the term MTB in the very first instance.

In the introduction section, lines 81-84, authors should move the contents to the next paragraph starting from "The purpose of the present study".

Please write the scientific names in Italicize throughout the manuscript, e.g. line 106 and many others.

In line 111, please remove the words "described previously by colleagues of our laboratory"

Authors should correct the presentation of the scientific units throughout the manuscript

In Figure 1, Why authors have not taken as adjuvant and PBS as control. Please discuss in the results section.

In figure 2, Authors should mention the size of gene, cumulative size of 5 genes. What is the insert size of pET expression vector? Please discuss in the results section.

Figure 5, Role of BCG is missing in figure 5a, and figure 5b? Authors should include BCG effects on interferon secretion.

Author Response

Response to Reviewer 2 Comments

Point 1: In the introduction section, lines 30, authors should expand the term MTB in the very first instance.

Response 1: Thank you for your suggestion, the full name of the term MTB was added in line 13.

Point 2: In the introduction section, lines 81-84, authors should move the contents to the next paragraph starting from "The purpose of the present study".

Response 2: Thank you for your suggestion, the contents in lines 81-84 have been moved to the next paragraph and adjusted accordingly (lines 85-90).

Point 3: Please write the scientific names in Italicize throughout the manuscript, e.g. line 106 and many others.

Response 3: Thank you for your advice, the gene names and digestive enzyme names have been changed in italics (lines 105-115).

Point 4: In line 111, please remove the words "described previously by colleagues of our laboratory." Authors should correct the presentation of the scientific units throughout the manuscript.

Response 4: Thank you for your suggestion, the words "described previously by colleagues of our laboratory" have been removed and the scientific units throughout the manuscript have been checked.

Point 5: In Figure 1, Why authors have not taken as adjuvant and PBS as control. Please discuss in the results section.

Response 5: In the present study, mice immunized with adjuvant and PBS were taken as negative and blank controls respectively, part â‘¡ of figure 1 (line 328) exhibited the immunization strategy and 5 experimental groups including Epera013f, Epera013m, BCG, PBS and adjuvant groups.

Point 6: In figure 2, Authors should mention the size of gene, cumulative size of 5 genes. What is the insert size of pET expression vector? Please discuss in the results section.

Response 6: Thank you for your suggestion, the inserted gene size of Epera013f and 5 antigen components of Epera013m in pET expression vectors have been added in the results section (lines 244-245)

Point 7: Figure 5, Role of BCG is missing in figure 5a, and figure 5b? Authors should include BCG effects on interferon secretion.

Response 7: Thank you for your question, Figure 5A/B (line 426) showed the secreting capacity of the splenocytes from Epera013f and Epera013m immunized mice in response of five single antigen components stimulation, the results of Figure 5A/B demonstrated that all five individual antigens were able to induce increased Th1 and Th2 immune response in mice from Epera013f and Epera013m groups.

Figure 5 C/D showed BCG effects on IFN-γ and IL-4 secretion.

Round 2

Reviewer 2 Report

N.A.

Author Response

Thank you for changing the general comments  on our manuscript.   : )